# A GENERATIVE MODEL FOR GAME THEORY WITH FLOW EQUILIBRIUM

## ABSTRACT

In recent years, generative models have emerged as a groundbreaking development in the field of artificial intelligence, transforming various domains such as image synthesis, natural language processing, and data generation. While recent studies have integrated generative models into multi-agent scenarios, their game-theoretical implications have remained largely unexplored. Specifically, the relationship between solutions derived from generative models and game theoretical equilibrium concepts lacks rigorous investigation. This paper aims to bridge the gap between generative models and game theory by introducing a novel probabilistic framework for modelling multi-agent decision-making problems. This innovative framework reinterprets these problems as generative processes. Furthermore, we introduce a training objective known as "flow equilibrium" and establish a theoretical connection between flow equilibrium and Nash equilibrium. To analyse the theoretical properties of our framework, we present a tabular version algorithm along with a convergence proof. Additionally, we propose an extended algorithm incorporating neural networks to handle more complex environments. Notably, our framework naturally incorporates opponent modelling. Harnessing the capabilities of generative models, our framework excels in capturing the dynamics of strategic interactions among agents. We validate our approach through testing on various multi-agent tasks, including cooperative and general-sum games. The empirical results consistently support our theoretical findings, demonstrating that our framework consistently outperforms existing methods in terms of solution quality.

## 1 INTRODUCTION

In recent years, the field of artificial intelligence has witnessed remarkable advancements, primarily driven by the emergence of generative models. These models have brought about transformative changes across various domains, ranging from the generation of highly realistic images to the enhancement of natural language understanding (OpenAI, 2023) and the generation of diverse data (Ramesh et al., 2022). Their exceptional capacity to capture complex data distributions has made them a cornerstone of contemporary AI research. Recent studies have been inspired to harness the potent capabilities of generative models for addressing decision-making problems (Janner et al., 2022; Ajay et al., 2023; Lu et al., 2023; Liang et al., 2023). These methods solve the decision making problem by casting it as a generative process. They achieve this by recasting decision-making as a generative process, where the policy is represented by a generative model. These efforts have underscored the effectiveness of generative modelling in tackling sequential decision-making challenges, as evidenced by empirical results. While some researchers have extended the application of generative models to multi-agent decision-making problems (Zhu et al., 2023; Li et al., 2023), their focus has predominantly been on offline settings. Moreover, the game-theoretical analysis of policies derived from generative models has largely remained unexplored. Control as Inference (CAI) (Levine, 2018), rooted in probabilistic inference and the maximisation of likelihood, faces inherent challenges when applied directly to multi-agent decision-making. In such scenarios, agents often pursue multi-dimensional objectives that may not align with each other. Self-interest can lead to conflicts within the objective functions, complicating the optimisation of these probabilistic models.

This paper endeavours to bridge the theoretical gap between generative modelling and game theory by proposing a novel generative framework. Within this framework, we introduce a specialised prob-

abilistic process to model agent interactions, effectively transforming multi-agent decision-making problems into generative processes amenable to online interaction with generative models. We introduce a training objective named "flow equilibrium" for the generative model and establish a theoretical connection between flow equilibrium and Nash equilibrium. Building upon this framework, we present a tabular version algorithm along with a convergence proof. Furthermore, we propose a parameterised algorithm that incorporates neural networks, extending its applicability to complex environments. Additionally, we seamlessly integrate opponent modelling into our framework. Leveraging the capabilities of generative models, our framework excels in capturing the dynamics of strategic interactions among agents, with an analysis of error bounds for opponent modelling. We evaluate the performance of our framework in differential games and non-atomic routing games against strong baseline methods, demonstrating its superior overall performance.

## 2 RELATED WORKS

**Generative Model**    A generative model is a type of statistical model that is designed to generate or produce new data samples. Generative models learn the underlying structure or patterns in the training data and then generate new data points by sampling from the learned distribution. Autoregressive models (Larochelle & Murray, 2011; Germain et al., 2015), normalising flow models (Dinh et al., 2015), and variational auto-encoders (VAEs) (Kingma & Welling, 2014; Rezende et al., 2014) directly learn the distribution's probability function via maximum likelihood while generative adversarial networks (GANs) (Goodfellow et al., 2014) represent the probability distribution implicitly by a model of its sampling process. Recent Large Language Models (LLMs) such as GPT-4 (OpenAI, 2023) have demonstrated their remarkable language prowess. In the field of image synthesis, diffusion models have displayed their outperforming abilities (Song et al., 2021). In this paper, we propose a new generative model for solving game theory, extending the application of generative model.

**Opponent Modelling**    In multi-agent reinforcement learning (MARL), learning a robust policy against the uncertainty caused by the unknown opponent policy is crucial. To mitigate this uncertainty, opponent modelling aims to model the opponent's behaviours, goals, or beliefs, thereby reducing the uncertainty. One line of the work is to predict the opponent behaviour using imitation learning (Grover et al., 2018). ToMnet leverages Theory of Mind to infer the agent's actions and goals from past and current observations (Rabinowitz et al., 2018). SOM explicitly model opponent using an agent's own policy to predict an opponent's action based on the opponent's state. The agent then use gradient descent to optimise its belief about the opponent's goal. PR2 (Wen et al., 2019b) and GR2 (Wen et al., 2019a) employ the recursive reasoning based on the joint Q function which requires extra information. We solve the opponent model leveraging the powerful generative model.

**Control as Inference**    Variational inference (VI) is a powerful tool to learn and inference probabilistic models (Jordan et al., 1999; Zhang et al., 2018). VI works by approximating the target distribution through the minimisation of a divergence objective. Casting a control problem into a probability inference problem enables the application of advanced inference tools to the control, and extends the model of control. Applying probabilistic inference to control has a long history (Toussaint, 2009a;b; Rawlik et al., 2010; 2013; Toussaint & Storkey, 2006), (Dvijotham & Todorov, 2012). Casting a control problem into a probability inference problem enables the application of advanced inference tools to the control, and extends the model of control (Levine, 2018; Kappen et al., 2009). However, most of the existing works focus on the single-agent case. There are a few works that try to extend the inference framework to the multi-agent setting, due to interest conflicts among agents. And most of them focus on cooperative games (Tian et al., 2019; Wen et al., 2020), which limits the application of the framework.

## 3 PRELIMINARIES

### 3.1 MARKOV DECISION PROCESS

We consider a Markov decision process (MDP) with $N$ agents. An MDP is characterised by a tuple $\mathcal{M} = (\mathcal{S}, \mathcal{N}, \{\mathcal{A}_i, r_i\}_{i \in \mathcal{N}}, P, \mu_0, \gamma)$. $\mathcal{S}$ is a finite state space. $\mathcal{N} = \{1, 2, \ldots, N\}$ is the set of agents. $\mathcal{A}_i$ is the action space of agent $i$. $\mathcal{A} = \times_i \mathcal{A}_i$ is the space of joint action space. $r_i : \mathcal{S} \times \mathcal{A} \to \mathbb{R}$ is the

reward function of agent $i$. $P : \mathcal{S} \times \mathcal{A} \to \mathcal{P}(\mathcal{S})$ is transition kernel for the state dynamic. $\mu_0$ is the initial distribution of initial state $s_0$. $\gamma \in (0, 1)$ is the discount factor for future rewards. The MDPs consider only one self-interested players, which limits the its flexibility in modelling the uncertainty in the external environment.

## 3.2 STOCHASTIC GAME

The stochastic game (SG) extends MDPs to accommodate scenarios involving multiple self-interested players. We consider a SG (Shapley, 1953; Shoham & Leyton-Brown, 2008) with $N$ players. The horizon of SG is $\mathcal{T} = \{0, 1, \ldots, T\}$. At each time index $t \in \mathcal{T}$, agent $i \in \mathcal{N}$ ($\mathcal{N} = \{1, 2, \ldots, N\}$) at state $s_t \in \mathcal{S}$ will select an action $a_t^i$ from the action space $\mathcal{A}^i$. All the agents take action simultaneously. Let $\boldsymbol{a}_t = (a_t^1, a_t^2, \ldots, a_t^N) \in \mathcal{A}$ denote the joint action. Each agent $i$ will receive a reward $r^i(s_t, \boldsymbol{a}_t)$ and the joint state will change to $s_{t+1}$ according to the transition kernel $P(s_{t+1}|s_t, \boldsymbol{a}_t)$. Agents $i$ take actions according to policy $\pi^i : \mathcal{S} \to \Delta(\mathcal{A}^i)$. Given the joint policy $\boldsymbol{\pi} = (\pi_1, \pi_2, \ldots, \pi_N)$, the cumulative reward of agent $i$ is

$$V^i(s; \boldsymbol{\pi}) = \sum_{t=0}^{\infty} \mathbb{E} \left[ \gamma^t r_t^i(s_t, \boldsymbol{a}_t)|s_0 = s, \boldsymbol{\pi} \right], \tag{1}$$

where the expectation is taken with respect to $s_{t+1} \sim P(\cdot|s_t, \boldsymbol{a}_t)$, $\boldsymbol{a}_t \sim \boldsymbol{\pi}(\cdot|\boldsymbol{s}_t)$. The Nash equilibrium is a joint policy $\boldsymbol{\pi}^* = (\pi^{1,*}, \pi^{2,*}, \ldots, \pi^{N,*})$ such that for all agent $i$, $V^i(s; \boldsymbol{\pi}^*) \geq V^i(s; \pi^i, \boldsymbol{\pi}_{-i}^*)$, where $\boldsymbol{\pi}^{-i,*} = (\pi^{1,*}, \ldots, \pi^{i-1,*}, \pi^{i+1,*}, \ldots, \pi^{N,*})$, i.e. $\pi^{i,*}$ is the best response of $\boldsymbol{\pi}^{-i,*}$. Accordingly, $\pi^{i,*} \in \Pi^i := \Delta(\mathcal{A}_i)$, $\boldsymbol{\pi}^* \in \Pi := \Delta(\times_{i \in \mathcal{N}} \mathcal{A}_i)$ and $\boldsymbol{\pi}^{-i,*} \in \Pi := \Delta(\times_{i \neq j \in \mathcal{N}} \mathcal{A}_j)$. Similarly, a joint policy $\boldsymbol{\pi}^*$ is the $\epsilon$-Nash equilibrium if there exists an $\epsilon > 0$ so that for all agent $i \in \mathcal{N}$, $V^i(s; \boldsymbol{\pi}^*) \geq \max_{\pi^i \in \Pi^i} V^i(s; \pi^i, \boldsymbol{\pi}^{-i,*}) - \epsilon$.

# 4 GRAPHICAL MODEL FOR GAME (GMG)

In this section, we first establish the an abstract graphical model for game from the probabilistic perspective and propose an equilibrium concept named flow equilibrium. Then we connect it with the game theory. We focus on a directed acyclic graph (DAG), represented by $\mathcal{G} = (\mathcal{X}, \mathcal{E})$, where $\mathcal{X}$ denotes a finite set of vertices, and $\mathcal{E} \subset \mathcal{X} \times \mathcal{X}$ refers to a set of directed edges.

We define a parent-child relationship between two vertices $x_i$ and $x_{i+1}$ when the directed edge $x_i \to x_{i+1}$ represents an action. Specifically, $x_i$ is the parent vertex of $x_{i+1}$, and $x_{i+1}$ is the child vertex of $x_i$. Furthermore, we define the *initial vertex* $x_1$ as the unique state with no incoming edges, and we refer to vertices that have no outgoing edges as *terminating vertices*.

Given the current vertex $x_t$, agent $i \in \mathcal{N} = \{1, 2, \cdots, N\}$ will sample action $a_t^i$ from the policy $\pi_t^i$. The joint action and the joint policy are denote as $\boldsymbol{a}_t = \{a_t^1, a_t^2, \cdots, a_t^N\}$ and $\boldsymbol{\pi}_t = \{\pi_t^1, \pi_t^2, \cdots, \pi_t^N\}$, respectively. Then the next vertex $x_{t+1}$ will be sampled from a fixed transition probability function $P(x_{t+1}|x_t, \boldsymbol{a}_t)$. A trajectory can be obtained by sampling states from policy $\boldsymbol{\pi}$ and transition probability $P$ successively. The probability to generate the trajectory $\tau$ is denote as $P(\tau; \boldsymbol{\pi})$. The marginal probability of sampling trajectories that ended at $x_T$ is given by $P_T(x_T; \boldsymbol{\pi}) = \sum_{\tau \to x_T} P(\tau; \boldsymbol{\pi})$, where $\tau \to x_T$ is defined as the set of trajectories that reach the terminating vertex $x_T$. When the terminating vertex $x_T$ is sampled, agent $i \in \mathcal{N}$ will receive a non-negative return function $R^i(x_T; \boldsymbol{\pi})$.

## 4.1 FLOW EQUILIBRIUM

We denote $\boldsymbol{\pi}^{-i}$ as the joint policy of all agents except $i$. Given the $\boldsymbol{\pi}^{-i}$, agent $i$ aims to update $\pi^i$ such that $P_T(x_T; \pi^i, \boldsymbol{\pi}^{-i}) \propto R^i(x_T; \boldsymbol{\pi})$. The goal to optimise the policies is to reaching an equilibrium named flow equilibrium (FE), which is defined as follows.

**Definition 4.1.** The flow equilibrium is a profile $\boldsymbol{\pi}^\star$ that satisfies the condition $P_T(x_T; \pi^{i,\star}, \boldsymbol{\pi}^{-i,\star}) \propto R^i(x_T; \boldsymbol{\pi})$ for all $i \in \mathcal{N}$ and any policy $\pi^i$, where $\boldsymbol{\pi}^{-i,\star}$ denotes the policy profile of all agents except $i$.

We prove that the FE exists as shown in the Theorem 4.2.

**Theorem 4.2.** *Given a non-negative function $R(x) = \{R^i(x_T; \boldsymbol{\pi})\}_{i \in \mathcal{N}}$ are continuous with respect to $\boldsymbol{\pi}$ and transition probability $P$, there exists an FE.*

## 4.2 Solving Markov Game

In this section, we will introduce how to use GMG to solve Markov game.

In the GMG, the return function $R(x)$ depends solely on the current vertex $x$, while in a Markov game, the objective is to maximise the long-term return as a function of a trajectory. Consequently, it should be able to determine the long-term return of a trajectory in a GMG using the terminal vertex $x_T$. A nature way to solve the problem is to choose the full trajectory in the Markov game as the vertex in GMG, but it will make the space complexity grows exponentially.

By utilising state augmentation, we establish a relationship between the vertex in GMG and a trajectory in the Markov game. For a given trajectory $\tau = (s_0, \boldsymbol{a}_0, s_1, \boldsymbol{a}_1, \cdots, s_T, \boldsymbol{a}_T)$, the accumulated reward up to time step $t$ is denoted as $z_t(\tau) = \frac{\sum_{k=0}^{t-1} \gamma^k r^i(s_k, \boldsymbol{a}_k)}{\gamma^{t-1}}$. As this equation is true for each agent $i$, we omit the $i$ for brevity. We select the vertex in the GMG as $x_t = (s_t, z_t, p_t)$. The transition functions can be expressed as follows:

$$s_{t+1} \sim P(\cdot|s_t, \boldsymbol{a}_t)$$
$$z_{t+1} = r^i(s_t, \boldsymbol{a}_t) + \frac{z_t}{\gamma}$$
$$p_{t+1} = p_t \boldsymbol{\pi}^{-i}(\boldsymbol{a}_t^{-i}|s_t) P(s_{t+1}|s_t, \boldsymbol{a}_t).$$

This vertex choice enable us to compute the long-term return from the terminating vertex, which reduces space complexity in comparison to choosing the entire trajectory as the vertex. The return function of GMG is non-negative, so we choose $R^i(x_T) = \exp(p_T \sum_{t=0}^{\infty} \gamma^t r^i(s_t, a_t)) = \exp(\text{Ret}^i(\tau))$.

The next question to apply GMG to solving Markov game is the connection between FE and NE.

**Theorem 4.3.** *If $\boldsymbol{\pi}$ is an FE, it is an $\epsilon$-NE with $\epsilon = 2|\mathcal{X}| Ret_{max} e^{-\delta}$, where $|\mathcal{X}| = \max_i |\mathcal{X}^i|$, $Ret_{max} = \max_{\tau, i} Ret^i(\tau)$, and $\delta = \min_i Ret_{max} - \max_{Ret^i(\tau) < Ret_{max}} Ret^i(\tau)$.*

The proof is deferred to Appendix A.2.

## 4.3 Training Criterion

In this section, we employ variational inference to solve Markov game under the framework of GMG. To achieve flow equilibrium, we want to minimise the KL divergence $\text{KL}(P_T(x_T; \pi^i, \boldsymbol{\pi}^{-i}) \| R^i(x_T)/Z)$. From the convexity, we can optimise the upper bound of $\text{KL}(P_T(x_T; \pi^i, \boldsymbol{\pi}^{-i}) \| R^i(x_T)/Z)$.

$$\text{KL}(P_T(x_T; \pi^i, \boldsymbol{\pi}^{-i}) \| \mathbb{E}_{\pi^i, \boldsymbol{\pi}^{-i}}[R^i(x_T)]/Z) \leq \text{KL}(P(\tau; \boldsymbol{\pi}) \| R^i(x_T)/Z)$$

$$= \mathbb{E}_{\boldsymbol{\pi}, P}\left[ \log \frac{\prod_{t=1}^{\infty} \pi^i(a_t|s_t, \boldsymbol{a}^{-i}) \rho(\boldsymbol{a}^{-i}|s_t)}{\prod_{t=1}^{\infty} \boldsymbol{\pi}^{-i}(\boldsymbol{a}_t^{-i}|s_t)} - \sum_{t=0}^{\infty} \gamma^t r^i(s_t, \boldsymbol{a}_t) + \log Z \right]$$

$$= \mathbb{E}_{\boldsymbol{\pi}, P}\left[ -\sum_{t=0}^{\infty} \gamma^t r^i(s_t, \boldsymbol{a}_t) - \sum_{t=0}^{\infty} \gamma^t H(\pi^i(a_t^i|s_t, \boldsymbol{a}_t^{-i})) \right]$$

$$+ \mathbb{E}_{\boldsymbol{\pi}, P}\left[ \sum_{t=0}^{\infty} \gamma^t \text{KL}(\rho(\boldsymbol{a}^{-i}|s_t) \| \boldsymbol{\pi}^{-i}(\boldsymbol{a}_t^{-i}|s_t)) + \log Z \right], \tag{2}$$

where $\pi^i(a_t^i|s_t, \boldsymbol{a}_t^{-i})$ is the policy of the agent $i$ and $\rho(\boldsymbol{a}_t^{-i}|s_t)$ is the opponent model of agent $i$. It is worth emphasising that the minimisation of Equation 2 necessitates that updates to both the policy and the opponent model can only be performed upon the completion of the entire trajectory. To enhance sample efficiency, we define the action-value function and value function allowing to optimise the policy and opponent model through partial trajectories.

**Definition 4.4.** Given a joint policy $\boldsymbol{\pi}$, the action-value function is defined as follows.

$$Q^i(s_t, a_t^i, a_t^{-i}; \boldsymbol{\pi}) = r_t^i(s_t, \boldsymbol{a}_t) + \log \hat{\boldsymbol{\pi}}^{-i}(\boldsymbol{a}_t^{-i}|s_t)$$
$$+ \mathbb{E}\left[\sum_{k=t+1}^{\infty} \gamma^{k-t}(r_k^i(s_k, \boldsymbol{a}_k) + H(\pi^i(a_k^i|s_k, \boldsymbol{a}_k^{-i})) - \mathrm{KL}(\rho(a_k^{-i}|s_k)||\hat{\boldsymbol{\pi}}^{-i}(\boldsymbol{a}_k^{-i}|s_k)))\right], \quad (3)$$

where the expectation is taken with respect to $a_k^i \sim \pi^i(\cdot|s_k, \boldsymbol{a}_k^{-i})$, $\boldsymbol{a}_k^{-i} \sim \rho(\cdot|s_k)$, $s_{k+1} \sim P(\cdot|s_k, a_k^i, \boldsymbol{a}_k^{-i})$. And the value function is

$$V^i(s; \boldsymbol{\pi}) = \mathbb{E}[Q^i(s, a^i, \boldsymbol{a}^{-i}; \boldsymbol{\pi}) - \log \pi^i(a^i|s, \boldsymbol{a}^{-i})\rho(\boldsymbol{a}_t^{-i}|s_t)],$$

where the expectation is taken with respect to $a^i \sim \pi^i(\cdot|s, \boldsymbol{a}^{-i})$, $\boldsymbol{a}^{-i} \sim \rho(\cdot|s)$.

Leveraging the notation of action-value function and value function, we derive an equivalent form to minimise the upper bound in the Equation (2).

$$J^i(\pi^i, s_0; \boldsymbol{\pi}^{-i}) = \mathbb{E}_{s_0 \sim P(s_0)}\left[V^i(s_0; \boldsymbol{\pi})\right]$$
$$= \mathbb{E}_{s_0 \sim P(s_0)}\left[\mathbb{E}[Q^i(s_0, a^i, \boldsymbol{a}^{-i}; \boldsymbol{\pi}) - \log \pi^i(a^i|s_0, \boldsymbol{a}^{-i})] - \log \rho(\boldsymbol{a}^{-i}|s_0)\right]$$
$$= \mathbb{E}_{s_0 \sim P(s_0)}\left[\log Z + H(\rho(\cdot|s_0)) - \mathbb{E}\left[\mathrm{KL}\left(\pi^i(a^i|s_0, \boldsymbol{a}^{-i})\left\|\frac{\exp(Q^i(s_0, a^i, \boldsymbol{a}^{-i}; \boldsymbol{\pi}))}{Z}\right)\right]\right],$$

where $Z = \sum_{a^i \in A_i} \exp(Q^i(s_0, a^i, \boldsymbol{a}^{-i}; \boldsymbol{\pi}))$. Since the KL divergence is non-negative, we have the following proposition.

**Proposition 4.5.** *The best response policy is in the form of*

$$\pi^{i,*}(a^i|s, \boldsymbol{a}^{-i}) = \frac{\exp(Q^i(s, a^i, \boldsymbol{a}^{-i}; \boldsymbol{\pi}))}{\sum_{a^i \in A_i} \exp(Q^i(s, a^i, \boldsymbol{a}^{-i}; \boldsymbol{\pi}))}. \quad (4)$$

Note that action-value function defined here differs from the Q function in the context of reinforcement learning. The expectation of action-value function is taken with respect to the opponent model while the expectation of Q function in the context of reinforcement learning is taken with respect to the opponent policy $\boldsymbol{\pi}^{-i}$. Therefore, the action value function is not the expected cumulative reward. The following proposition provides the upper bound for this difference.

**Proposition 4.6.** *Suppose that* $\mathrm{KL}(\rho(\cdot|s)||\boldsymbol{\pi}^{-i}(\cdot|s)) < \epsilon_\rho$ *for all* $s \in \mathcal{S}$. *Without loss of generality, the reward function* $|r^i(s, \boldsymbol{a})| \leq 1$, $\forall s \in \mathcal{S}$, $\boldsymbol{a} \in \mathcal{A}$, $i \in \mathcal{N}$. *Denote the action-value function derived using the opponent model as* $\hat{Q}^i(s, \boldsymbol{a}; \boldsymbol{\pi})$. *Then we have that*

$$\max_{s \in \mathcal{S}, \boldsymbol{a} \in \mathcal{A}, i \in \mathcal{N}} |Q^i(s, \boldsymbol{a}; \boldsymbol{\pi}) - \hat{Q}^i(s, \boldsymbol{a}; \boldsymbol{\pi})| \leq \delta, \quad (5)$$

*where* $\delta := \frac{2(1+\log|\mathcal{A}_i|)}{(1-\gamma)^2}\sqrt{\frac{1}{2}\epsilon_\rho} + \frac{\epsilon_\rho}{1-\gamma}$.

The proof is deferred to Appendix A.4. Note that the definition of the action-value function requires that we approximate the policy of opponents $\boldsymbol{\pi}^{-i}(\boldsymbol{a}_t^{-i}|s_t)$ with agent $i$'s opponent model $\rho(\boldsymbol{a}_t^{-i}|s_t)$. Here we don't specify the method to update $\rho(\boldsymbol{a}_t^{-i}|s_t)$. The above conclusion applies to any opponent model method.

In order to capture dynamics among agents, we propose an opponent modelling method under our framework. Here we consider the case $|\mathcal{N}| = 2$ for the brevity of notation, but this method can be extended to the cases with more agents. To approximate the behaviour of agent $-i$, we factorise the auxiliary distribution over states and actions $q(\boldsymbol{a}_{0:\infty}, s_{0:\infty})$ in the following way.

$$q(\boldsymbol{a}_{0:\infty}, s_{0:\infty}) = P(s_0)\prod_{t=0}^{\infty} q(\boldsymbol{a}_t^{-i}|s_t)q(a_t^i|s_t, \boldsymbol{a}_t^{-i})P(s_{t+1}|s_t, \boldsymbol{a}_t^{-i}, a_t^i)$$

$$= P(s_0)\prod_{t=0}^{\infty} \rho(\boldsymbol{a}_t^{-i}|s_t)\pi^i(da_t^{-i}|s_t, \boldsymbol{a}_t^{-i})P(s_{t+1}|s_t, \boldsymbol{a}_t^{-i}, a_t^i)$$

We denote the solution to this problem as the joint policy $\boldsymbol{\pi}^*$. Denote $Q_\rho^{-i}(s_t, \boldsymbol{a}_t; \rho)$ as the soft action-value function of agent $-i$.

$$
\begin{aligned}
Q_\rho^{-i}(s_t, \boldsymbol{a}_t; \rho) = & r^{-i}(s_t, \boldsymbol{a}_t) - \mathrm{KL}(\hat{\boldsymbol{\pi}}^{-i}(\cdot|s_t)\|\rho(\cdot|s_t)) \\
& + \mathbb{E}[\sum_{h=t+1}^\infty \gamma^{h-t}(r_h^{-i}(s_h, \boldsymbol{a}_h) - \mathrm{KL}(\hat{\boldsymbol{\pi}}^{-i}(\cdot|s_h)\|\rho(\cdot|s_h)))],
\end{aligned}
$$

where the expectation is taken with respect to $\boldsymbol{a}_h \sim q(\cdot|s_h)$, $s_h \sim P(\cdot|s_{h-1}, \boldsymbol{a}_{h-1})$. $\hat{\boldsymbol{\pi}}^{-i}$ is the empirical distribution of opponent policy. Then we can derive the optimal opponent model for agent $-i$.

**Proposition 4.7.** *The optimal opponent model for agent $i$ is*

$$
\rho(\boldsymbol{a}^{-i}|s) = \frac{\hat{\boldsymbol{\pi}}^{-i}(\boldsymbol{a}^{-i}|s)\exp(\mathbb{E}_{a^i\sim\pi^i}[Q_\rho^{-i}(s, \boldsymbol{a}; \rho)])}{\mathbb{E}_{\boldsymbol{a}^{-i}\sim\hat{\boldsymbol{\pi}}^{-i}(\cdot|s)}\left[\exp(\mathbb{E}_{a^i\sim\pi^i}[Q_\rho^{-i}(s, \boldsymbol{a}; \rho)])\right]} \tag{6}
$$

*where $\hat{\boldsymbol{\pi}}^{-i}(\boldsymbol{a}^{-i}|s)$ is the prior of opponent policy $\boldsymbol{\pi}^{-i}(\boldsymbol{a}^{-i}|s)$.*

The proof is deferred to Appendix A.8. Proposition 4.7 provides a closed-form of opponent modelling. Note that the result resembles a logit quantal response equilibrium (LQRE) policy when the prior of the opponent's policy is a uniform distribution. This finding suggests that our opponent modelling framework is well-suited for handling uncertainty from the external environment.

# 5 GENERAL VARIATIONAL BAYESIAN OPPONENT MODELLING

In this section, we aim to propose an algorithm to solve the Markov game based on our framework and training criterion. Then we prove that this algorithm converges on the Markov Potential Game (MPG). Further, we propose GPI, an actor critic algorithm powered by neural networks to solve complex and continuous problem.

## 5.1 VARIATIONAL POLICY GRADIENT

Proposition 4.5 shows that the best response policy is in the form of $\pi^{i,\theta}(a|s, \boldsymbol{a}^{-i}) = \mathrm{softmax}(\theta_{i,s,a,\boldsymbol{a}^{-i}})$, i.e. the best response policy is softmax policy parameterised (Agarwal et al., 2021). We use the natural policy gradient (NPG) method (Kakade, 2001) to derive the best response policy.

**Proposition 5.1.** *Denote $\theta^{(t)}$ the $t$-th iterate and $\pi^{(t)} = \mathrm{softmax}(\theta_{s,\boldsymbol{a}})$. For each agent $i$, state $s$, and action $a$, the NPG update rule can be written as*

$$
\pi^{i,(t+1)}(a\mid s, \boldsymbol{a}^{-i}) = \frac{1}{Z^{(t)}(s)}\left(\pi^{i,(t)}(a\mid s, \boldsymbol{a}^{-i})\right)^{1-\frac{\eta}{1-\gamma}}\exp\left(\frac{\eta Q^{i,(t)}(s, a, \boldsymbol{a}^{-i}; \boldsymbol{\pi}^{(t)})}{1-\gamma}\right). \tag{7}
$$

*where $\eta$ is the learning rate.*

The proof is deferred to Appendix A.3. Then we propose the variational policy gradient (VPG) algorithm. The pseudo-code of VPG is listed in the Algorithm 1.

Then we will prove that VPG converges to Nash equilibrium in the Markov potential game.

**Definition 5.2.** MPG is a Markov decision process that there exists a function $\Phi(s; \pi^i, \boldsymbol{\pi}^{-i}) : \Pi \to \mathbb{R}$, with $s \in \mathcal{S}$, so that

$$
\tilde{V}^i\left(s; \pi^i, \boldsymbol{\pi}^{-i}\right) - \tilde{V}^i\left(s; \pi^{i,\prime}, \boldsymbol{\pi}^{-i}\right) = \Phi\left(s; \pi^i, \boldsymbol{\pi}^{-i}\right) - \Phi\left(s; \pi^{i,\prime}, \boldsymbol{\pi}^{-i}\right),
$$

for all agents $i \in \mathcal{N}$, states $s \in \mathcal{S}$ and policies $\pi^i, \pi^{i,\prime} \in \Pi^i, \boldsymbol{\pi}^{-i} \in \Pi_{-i}$. Here $\tilde{V}^i\left(s; \pi^i, \boldsymbol{\pi}^{-i}\right)$ is value function with accurate opponent modelling.

The first step is to prove that the estimation error of the opponent is bounded. The proposition 4.6 has provided the upper bound of the estimation error of the opponent.

---

**Algorithm 1** Variational Policy Gradient (VPG)

---

**input** Learning rate $\eta$
  Initialise opponent model $\rho$.
  Initialise policy $\pi^{i,(0)}$ for all agent $i \in \mathcal{N}$.
  Initialise the replay buffer $M$.
  **for** $k = 1, 2, \ldots$ **do**
    **for** Each agent $i \in \mathcal{N}$ **do**
      For the current state $s_t$, $a_t^i \sim \pi^i(\cdot|s_t) = \sum_{\boldsymbol{a}_t^{-i}} \rho(\boldsymbol{a}_t^{-i}|s_t)\pi^i(\cdot|s_t, a_t^i, \boldsymbol{a}_t^{-i})$.
      Observe next state $s_{t+1}$, opponent action $a_t^{-i}$ and reward $r_t^i$ and save the experience in the reply buffer.
      Update opponent model.
    **end for**
    **for** Each agent $i \in \mathcal{N}$ **do**
      Compute the best response policy using Equation (7).
    **end for**
  **end for**

---

The second step is to derive the convergence of VPG with exact opponent modelling. We first show the equivalence between VPG and the global NPG on the potential function. Then we will prove the convergence of VPG using the smoothness of the potential function.

Note that the gradient of the value functions equals the potential function and agents update their policy independently. Hence VPG is equivalent to running Natural Policy Gradient (NPG) on the potential function, which is shown in the following proposition.

**Proposition 5.3.** *Consider the global NPG dynamic on the potential function:* $\theta_s^{(t+1)} = \theta_s^{(t)} + \eta \mathcal{F}^\dagger(\theta_s^{(t)})\nabla_{\theta_s}\Phi \; \forall s \in \mathcal{S}$, *where* $\mathcal{F}^\dagger(\theta_s) = \mathbb{E}[\nabla_{\theta_s}\log\boldsymbol{\pi}^{\theta_s}(\boldsymbol{a}|s)\nabla_{\theta_s}\log\boldsymbol{\pi}^{\theta_s}(\boldsymbol{a}|s)^T]^\dagger$ *is the pseudo-inverse of the Fischer information matrix.* $\boldsymbol{\pi}^{\theta_s}(\boldsymbol{a}|s) = \prod_{i\in\mathcal{N}}\mathbb{E}_{a^{-i}\sim\rho(\cdot|s)}[\pi^i(a^i|s, a^{-i})]$. *VPG has the same dynamics as global NPG.*

The proof is deferred to Appendix A.5. After showing the connection of VPG and the NPG on the potential function, we next show the smoothness of the potential function in the following lemma.

**Lemma 5.4.** *The potential function $\Phi$ is $L$-smooth with the constant* $L = \frac{2(n+1)^2}{(1-\gamma)^3} + 2(n^2 + n + 1)\frac{1+\log\max_{i\in\mathcal{N}}|A_i|}{(1-\gamma)^2} + \frac{3n+2}{1-\gamma}$.

The proof is deferred to Appendix A.6. Using Lemma 5.4, the potential function $\Phi(s; \boldsymbol{\pi}^{(t)})$ is non-decreasing if the learning rate is $\frac{1}{L}$ (Bubeck et al., 2015). We finally give the convergence of VPG.

**Theorem 5.5.** *VPG converges to a fixed point, which is $\epsilon$-Nash equilibrium of MPG, where* $\epsilon = \delta + \frac{\log|A|}{1-\gamma}$.

Theorem 5.5 ensures the applicability of VPG for solving MPG.

VPG does not involve a certain opponent modelling method. The next question is how to model the opponent using variational inference.

### 5.2 OPPONENT MODELLING IN GENERAL-SUM GAME

Since the agent $i$ does not know the reward of the agent $-i$, we have to find a function $\hat{r}^{-i}$ to estimate $r^{-i}$. The objective of optimising $\hat{r}^{-i}$ is to minimise the KL divergence between the optimal opponent model derived by estimated reward function $\hat{r}^{-i}$ and the history data of agent $-i$. Let $\tau_{-i} = \{s_0, \boldsymbol{a}_0^{-i}, a_0^i, s_1, \boldsymbol{a}_1^{-i}, a_1^i, \ldots\}$ be the historical interaction data. The probability of generating $\tau_{-i}$ by the opponent model is

$$\rho(\tau_{-i}) = P(s_0)\prod_{t=1}^{\infty} P(s_t|s_{t-1}, \boldsymbol{a}_{t-1}^{-i}, a_{t-1}^i)\rho(\boldsymbol{a}_{t-1}^{-i}|s_{t-1})\pi^i(a_{t-1}^i|s_{t-1}).$$

Then the objective to optimise $\hat{r}^{-i}$ is

$$\text{KL}(P(\tau_{-i})||\rho(\tau_{-i})) = \mathbb{E}\left[\sum_{t=0}^{\infty} -\gamma^t \hat{r}^{-i}(s_t, \boldsymbol{a}_t^{-i}, a_t^i)\right] + \log \mathbb{E}_{\boldsymbol{a}^{-i} \sim \rho}\left[\mathbb{E}_{a^i \sim \rho_i}[\exp(Q_\rho^{-i}(s, \boldsymbol{a}; \rho))]\right], \quad (8)$$

where the first expectation is taken with respect to $s_t \sim P(s_t|s_{t-1}, \boldsymbol{a}_{t-1}^{-i}, a_{t-1}^i)$. It is difficult to calculate the optimal opponent model because $\mathbb{E}_{\boldsymbol{a}^{-i} \sim \rho}\left[\exp(Q_\rho^{-i}(s, \boldsymbol{a}; \rho))\right]$ is difficult to estimate. We use a sample-based method for estimating $\mathbb{E}_{\boldsymbol{a}^{-i} \sim \rho}\left[\exp(Q_\rho^{-i}(s, \boldsymbol{a}; \rho))\right]$.

$$\text{KL}(P(\tau_{-i})||\rho(\tau_{-i})) = \mathbb{E}\left[\sum_{t=0}^{\infty} -\gamma^t \hat{r}^{-i}(s_t, \boldsymbol{a}_t^{-i}, a_t^i)\right] + \log \mathbb{E}_{\tau_{-i} \sim \rho(\tau_{-i})}\left[\frac{\exp(\sum_{t=0}^{\infty} \gamma^t \hat{r}^{-i}(s_t, \boldsymbol{a}_t))}{\rho(\tau_{-i})}\right] \quad (9)$$

If $\hat{r}_\psi^{-i}$ is parameterised by $\psi$, the gradient of $\text{KL}(P(\tau_{-i})||\rho(\tau_{-i}))$ with respect to $\psi$ is

$$\frac{\text{dKL}(P(\tau_{-i})||\rho(\tau_{-i}))}{\text{d}\psi} = \mathbb{E}\left[\sum_{t=0}^{\infty} -\gamma^t \frac{\text{d}\hat{r}_\psi^{-i}(s_t, \boldsymbol{a}_t^{-i}, a_t^i)}{\text{d}\psi}\right] + \frac{1}{Z}\mathbb{E}_{\tau_{-i} \sim \rho(\tau_{-i})}\left[w_{-i}\frac{\text{d}\sum_{t=0}^{\infty} \gamma^t \hat{r}_\psi^{-i}(s_t, \boldsymbol{a}_t)}{\text{d}\psi}\right], \quad (10)$$

where $w_{-i} = \frac{\exp(\sum_{t=0}^{\infty} \gamma^t \hat{r}_\psi^{-i}(s_t, \boldsymbol{a}_t))}{\rho(\tau_{-i})}$ and $Z = \mathbb{E}_{\tau_{-i} \sim \rho(\tau_{-i})}[w_{-i}]$.

VPG is for tabular cases and is impractical in problems with high dimensions or continuous action. To handle the problems, we propose the variational actor-critic method, which can be implemented in a complex continuous environment. We use neural-network to parameterise the policy $\pi^\theta$, opponent model $\rho^\phi$, the action-value function $Q_\omega$, and the reward function $r_\psi$.

The objective to optimise the policy $\pi^\theta$ is to minimise the KL divergence

$$J_\pi(\theta; s) = \mathbb{E}_{a^{-i} \sim \rho(\cdot|s)}\left[\text{KL}\left(\pi_i^\theta(\cdot|s)|| \exp(Q_\omega^i(s, \cdot, \boldsymbol{a}^{-i}) - V^i(s))\right)\right]. \quad (11)$$

The objective to optimise the action-value function $Q_\omega$ is to minimise:

$$J_Q(\omega) = \mathbb{E}_{(s_t, a_t^i, \boldsymbol{a}_t^{-i}) \sim \mathcal{D}}\left[\frac{1}{2}\left(Q_\omega^i(s_t, a_t^i, \boldsymbol{a}_t^{-i}) - r^i(s_t, a_t^i, \boldsymbol{a}_t^{-i}) - \gamma \mathbb{E}_{s_{t+1} \sim p_s}[\bar{V}(s_{t+1})]\right)^2\right], \quad (12)$$

with $\bar{V}^i(s_{t+1}) = Q_{\bar{\omega}}^i\left(s_{t+1}, a_{t+1}^i, \hat{\boldsymbol{a}}_{t+1}^{-i}\right) - \log \rho_\phi\left(\hat{\boldsymbol{a}}_{t+1}^{-i} \mid s_{t+1}\right) - \log \pi_\theta\left(a_{t+1}^i \mid s_{t+1}, \hat{\boldsymbol{a}}_{t+1}^{-i}\right) + \log \hat{\boldsymbol{\pi}}\left(\hat{\boldsymbol{a}}_{t+1}^{-i} \mid s_{t+1}\right)$, where $Q_{\bar{\omega}}^i$ is target action-value function.

The gradient of (11) with respect to $\theta$ is

$$\nabla_\theta J_\pi(\theta; s) = \mathbb{E}_{\boldsymbol{a}^{-i} \sim \rho(\cdot|s)}[\nabla_\theta \log \pi_i^\theta(a|s) + (\nabla_a \pi_i^\theta(a|a^i, s) - \nabla_a Q^i(s, a, \boldsymbol{a}^{-i}))\nabla_\theta f_\theta(\epsilon; s, \boldsymbol{a}^{-i})] \quad (13)$$

where $a$ is evaluated at $f_\theta(\epsilon; s, \boldsymbol{a}^{-i})$. The gradient of (12) with respect to $\omega$ is

$$\nabla_\omega J_Q(\omega) = \nabla_\omega Q_\omega^i(s_t, a_t^i, \boldsymbol{a}_t^{-i})\left(Q_\omega^i(s_t, a_t^i, \boldsymbol{a}_t^{-i}) - r^i(s_t, a_t^i, \boldsymbol{a}_t^{-i}) - \gamma \mathbb{E}_{s_{t+1} \sim p_s}[\bar{V}(s_{t+1})]\right) \quad (14)$$

Then the pseudo-code of the variational inference actor-critic method named Generative Policy Inference (GPI) is listed in the Algorithm 2.

## 6 EXPERIMENTS

As GPI incorporates entropy regularisation naturally, it enjoys stronger exploration ability. We test its exploration ability on a challenging differential game. Differential game is adopted from (Wei et al., 2018). The two agents in this game have continuous action space. All the agents share the same reward function depending on the joint action $(a_1, a_2)$ following the equations:
$r^1(a^1, a^2) = r^2(a^1, a^2) = \max(f_1, f_2)$, where $f_1 = 0.8 \times \left[-\left(\frac{a^1+5}{3}\right)^2 - \left(\frac{a^2+5}{3}\right)^2\right], f_2 = 1.0 \times \left[-\left(\frac{a^1-5}{1}\right)^2 - \left(\frac{a^2-5}{1}\right)^2\right] + 10$. The training process includes 200 episodes with 25 steps per episode. We compare GPI Deep Deterministic Policy Gradient (DDPG) (Lillicrap et al., 2016), and two multi-agent reinforcement learning algorithm: Multi-Agent Deep Deterministic Policy Gradient

---

**Algorithm 2** Generative Policy Inference (GPI)

Initialising replay buffer $\mathcal{D}$.
Initialising parameters $\theta$, $\omega$, $\psi$ and $\phi$.
**for** Each episode $d = 1, 2, \ldots$ **do**
    **for** $i \in \mathcal{N}$ **do**
        For current state $s_t$ compute $\boldsymbol{a}_t^{-i} \sim \rho(\cdot|s_t)$, $a_t^i \sim \pi^i(\cdot|s_t, a_t^{-i})$
        Observe next state $s_{t+1}$, opponent action $\boldsymbol{a}_t^{-i}$ and save the new experience in the reply buffer $\mathcal{D}$.
        Update opponent model using Algorithm 3.
        Update $\pi^i$ using Equation (13).
    **end for**
**end for**
**Output:** policy $\pi^i$, $i \in \mathcal{N}$, opponent model $\rho$

---

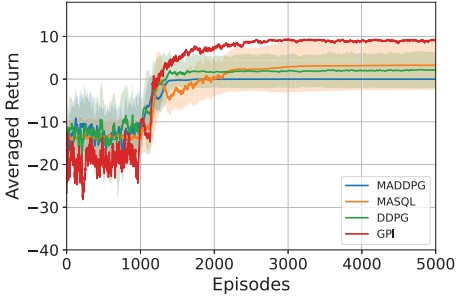

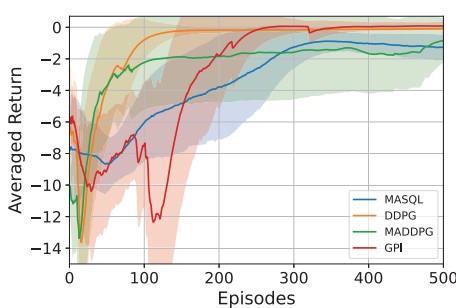

(a) The learning curves of GPI and other baselines in differential game.

(b) The learning curves of GPI and other baselines in non-atomic routing game.

Figure 1: Learning curves in differential game and non-atomic routing game.

(MADDPG) (Lowe et al., 2017), Multi-Agent Soft Q Learning (MASQL) (Wei et al., 2018) in this task. This is a challenging task for most continuous gradient-based RL algorithms because the gradient update often leads the training agent towards a suboptimal point. The reward surface has a local maximum of 0 at (-5, -5) and a global maximum of 10 at (5, 5), with a deep valley in between. If the agents' policies are initialized at (0, 0) (the red starred point), which is within the basin of the left local maximum, gradient-based methods are likely to struggle in reaching the global maximum equilibrium point because the valley blocks the upper right area. The learning curves are shown in Figure 1a. Only GPI shows the capability of converging to the global optimum, while other baselines can only reach the sub-optimal point.

To assess whether GPI can converge in a Markov potential game, we performed GPI on a task referred to as the non-atomic routing game. This game was borrowed from the work of Mguni et al. (Mguni et al., 2021). In this game, the agent and the virtual opponent are playing Markov potential game. Agents in this game are self-interested and learn how to split their commodity to maximise rewards. We compare GPI with MADDPG, MASQL and DDPG in this task. The learning curves are shown in Figure 1b. The learning curve of GPI is smoother and other algorithms suffer from high variance.

## 7 CONCLUSION

This paper bridges the gap between generative modelling and game theory in the field of artificial intelligence, recognising the transformative potential of generative models across various domains. By introducing a novel generative framework tailored to multi-agent decision-making scenarios and incorporating the concept of "flow equilibrium," we have addressed critical limitations and established theoretical connections to Nash equilibrium. Our proposed algorithms, including tabular and parameterised versions, combined with seamless opponent modelling, empower the field with versatile tools. Leveraging the expressive power of generative models, our framework excels in capturing dynamics among agents, as validated through empirical evaluations in differential and

non-atomic routing games where it consistently outperforms established baselines. This research not only fills a crucial void but also lays the groundwork for future advancements at the intersection of generative modelling and game theory.

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
