# OpenReview forum: "A Generative Model for Game Theory with Flow Equilibrium"
_ICLR.cc/2024/Conference — ICLR 2024 Conference Withdrawn Submission_

### Official Review · Reviewer_TMg4 · 2023-10-30

**Soundness:** 2 fair
**Presentation:** 2 fair
**Contribution:** 3 good
**Rating:** 3
**Confidence:** 3

**Summary:**

This paper frames learning a Nash equilibrium of stochastic game as minimizing the KL divergence between a generative model of agent play and the reward distribution of all agents over all end states at equilibrium. The generative model of agent play naturally includes models of each agent $i$'s individual policy as well as each agent $i$'s model of the remaining players' policies. The authors propose a variational policy gradient (VPG) update to learn this generative model (as well as a variational actor-critic method for more complex settings).

**Strengths:**

This paper proposes an interesting approach to equilibrium approximation in stochastic games. The idea of repurposing generative models for game theory is intriguing. Mapping the Nash equilibrium condition to a flow equilibrium and then attempting to minimize the distance between the visiting policy and the reward distribution is interesting as well.

**Weaknesses:**

I had a very hard time parsing some of the notation and derivations in this paper. I want to like this paper, but it appears to have been hastily written. Note it's also over the page limit. I have included more detailed questions below.

**Questions:**

- Theorem 4.2: You state that you define a mapping $\Gamma$ in Appendix A.1, but don't *actually* define $\Gamma$ anywhere. This is too sloppy given how central it is to your paper.
- Theorem 4.3: $\delta$ is defined with a $\max$ in the second term. What exactly is the $\max$ over? Trajectories? It's hard for me to understand the usefulness of this result if $\delta$ can be arbitrarily small (implying $\epsilon$ could be quite large). Note that $Ret_{\max}$ appears in Theorem 4.3 but is missing from the final claim in Appendix A.2 (equation 16).
- Equation 2: Have you defined $P$ in terms of $\pi$ explicitly anywhere? What about $R^i$ in terms of a $\gamma$ and $r^i$? Same with $Z$? It's difficult to follow the derivation here without knowing those terms. Why does a $\gamma^t$ appear in front of the entropy term $H$? Did you define $H$?
- Proposition 4.5: Please prove this result. I didn't see anything in the appendix to support it.
- Above equation 9, you say you use a sample-based method to estimate the expectation of the exponential of the Q values. Wouldn't Jensen's inequality say that the expectation of the exponential of the Q-values are an overestimate of the true value? How do you deal with this?
- What is equation 9 for? It seems to appear without any description in the text.

---

### Official Review · Reviewer_RfBQ · 2023-11-08

**Soundness:** 2 fair
**Presentation:** 1 poor
**Contribution:** 2 fair
**Rating:** 3
**Confidence:** 3

**Summary:**

This paper introduces a novel approach for addressing Markov decision processes (MDPs) involving multiple agents. The authors propose a graphical model representation of the game and define a concept called flow equilibrium. They establish a connection between flow equilibrium and the Nash equilibrium in Markov games and introduce a training criterion for solving the flow equilibrium. Leveraging this foundation, the authors propose variational policy gradient (VPG) and generative policy inference (GPI) methods for solving the Markov game and provide theoretical proofs of convergence for Markov potential games. The effectiveness of the approach is demonstrated through experiments on synthetic datasets.

**Strengths:**

1. The paper presents an interesting link between the proposed flow equilibrium and the Nash equilibrium in Markov games.

**Weaknesses:**

The paper is significantly hampered by its writing quality, which obscures the presentation and makes it challenging to assess the contributions. Specific issues include:

1. In Definition 4.1, $\pi^i$ is defined but is not utilized in subsequent equations, leading to confusion.
2. The proportionality $P\_T(x\_T; \pi^i, \pi^{-i}) \propto R^i(x\_T; \pi)$ is perplexing. $P\_T(x\_T; \pi) \propto R^i(x\_T; \pi)$ or $P\_T(x\_T; \pi^i, \pi^{-i}) \propto R^i(x\_T; \pi^i, \pi^{-i})$ would be better.
3. Theorem 4.2 confusingly states $R(x) = \{R^i(x\_T; \pi)\}$ when $R(x)$ should only be defined for terminating vertices.
4. The claim in Section 4.2 that "the return function $R(x)$ depends solely on the current vertex $x$" is misleading since $R(x)$ is previously defined only at terminating vertices.
5. Theorem 4.3 lacks clarity on what $\mathcal{X}^i$ represents, and the definition of GMG implies that all agents share the same vertex set.
6. In Theorem 4.3, there should be an assessment of $\epsilon$'s magnitude since a large $\epsilon$-NE could significantly deviate from a true NE.
7. Equation (2) introduces $\mathbb{E}\_{\pi^i, \pi^{-i}}[R^i(x\_T)]$ which is inconsistent with the notation $R^i(x\_T; \pi^i, \pi^{-i})$ used in Section 4.1.
8. The second term of Equation (2) is confusing as it attempts to calculate the KL divergence between distributions over different domains (trajectories vs terminating vertices).
9. The inclusion of $\rho$ in the third term of Equation (2) without it appearing in the first two terms is puzzling, raising questions about the derivation of the equation.
10. Definition 4.4 introduces $\hat{\pi}$ without an adequate explanation.
11. The derivations of Equation (3) and the subsequent equation between Equations (3) and (4) from Equation (2) are unclear.

Additional concerns extend beyond the mathematical content:
1. The paper does not elucidate the relationship between the VPG and GPI algorithms.
2. While the authors draw parallels between generative models and game theory, the connection to generative models within the proposed method is not clear.
3. The proposed methods are purported to be applicable to neural networks, yet no experimental validation is provided to substantiate their efficacy in such contexts.

In light of these points, the paper needs substantial revision to clarify its theoretical contributions and to provide a more robust empirical validation.

**Questions:**

See the weakness part.

---

### Official Review · Reviewer_39sv · 2023-11-09

**Soundness:** 3 good
**Presentation:** 3 good
**Contribution:** 3 good
**Rating:** 6
**Confidence:** 3

**Summary:**

The aim of this paper is to connect game theoretic views with generative models, by introducing a multi-agent decision framework that eventually generates the data. This is done by defining the flow equilibrium, which is a new solution concept where all $P_T$ values are in proportion to $R$. The manuscript further shows that such a solution concept must exist, and characterizes that they resemble logit quantal response equilibrium. Based on the concept, the manuscript proposes variational policy gradient, which finds the best response of the aforementioned opponent model. It shows that this variant of policy gradient will converge in Markov potential games. Some simple experiments follow.

**Strengths:**

1. This paper connects game theory and generative models, and propose a new solution concept of flow equilibrium. This idea is relatively new.
2. They show one use case of the solution concept, which is to model the opponent in multi-agent decision tasks. Such an opponent modeling is compatible with multi-agent learning methods such as policy gradient.
3. Some statements and experiments are provided.

**Weaknesses:**

I would expect such a solution concept to be a bit more "useful" to be more beneficial to the community. At the moment its use case is to improve the opponent modeling part, which plausibly introduces some marginal improvement to multi-agent decision algorithms (which agrees with the experiments).

**Questions:**

N/A